# RFS: Reinforcement Learning with Residual Flow Steering for Dexterous Manipulation

**Entong Su**[1], **Tyler Westenbroek**[1], **Anusha Nagabandi**[2], **Abhishek Gupta**[1]

[1]University of Washington, Paul G. Allen School of Computer Science & Engineering
[2]Amazon Frontier AI & Robotics

## Abstract

Imitation learning has emerged as an effective approach for bootstrapping sequential decision-making in robotics, achieving strong performance even in high-dimensional dexterous manipulation tasks. Recent behavior cloning methods further leverage expressive generative models, such as diffusion models and flow matching, to represent multimodal action distributions. However, policies pretrained in this manner often exhibit limited generalization and require additional fine-tuning to achieve robust performance at deployment time. Such adaptation must preserve the global exploration benefits of pretraining while enabling rapid correction of local execution errors. We propose Residual Flow Steering (RFS), a data-efficient reinforcement learning framework for adapting pretrained generative policies. RFS steers a pretrained flow-matching policy by jointly optimizing a residual action and a latent noise distribution, enabling complementary forms of exploration: local refinement through residual corrections and global exploration through latent-space modulation. This design allows efficient adaptation while retaining the expressive structure of the pretrained policy. We demonstrate the effectiveness of RFS on dexterous manipulation tasks, showing efficient fine-tuning both in simulation and in real-world settings when adapting pretrained base policies. Project website: https://weirdlabuw.github.io/rfs/

## 1 Introduction

Imitation learning from human demonstrations has proven effective for learning high-performing policies in robotics and sequential decision making (Zhao et al., 2023; 2025; Chi et al., 2023; Hussein et al., 2017; Ho and Ermon, 2016). Recent progress has been driven by *generative* imitation learning, which leverages expressive generative models such as diffusion models (Ho et al., 2020) and flow models (Lipman et al., 2022) to capture the distribution of human demonstrations. These approaches are particularly well suited to the inherent multimodality of human behavior (Chi et al., 2023; Black et al., 2024), enabling robust empirical scaling to complex manipulation tasks (Team et al., 2025). However, imitation learning alone often fails to generalize to all test-time scenarios (Team et al., 2025), leaving substantial headroom for policy improvement through test-time fine-tuning toward production-level performance.

While supervised fine-tuning (Ouyang et al., 2022; Black et al., 2024) has proven effective for decision-making systems, it depends on high-quality, curated expert data. Instead, we seek methods that avoid this dependency by leveraging cheaper, suboptimal, or self-collected data. Reinforcement learning (RL) and offline reinforcement learning (offline RL) (Levine et al., 2020) provide a principled framework for behavior improvement, as agents optimize reward objectives rather than relying on expert demonstrations. Although many methods have been proposed to combine imitation learning pretraining with reinforcement learning fine-tuning (Rajeswaran et al., 2018a; Hu et al., 2024; Nair et al., 2018; Nakamoto et al., 2024), these approaches typically rely on closed-form likelihoods (Lillicrap et al., 2015) and reparameterization (Wang et al., 2019), making them incompatible with expressive generative architectures such as diffusion and flow models. Moreover, a key challenge in this setting is balancing adaptation to new tasks with the preservation of knowledge acquired during pretraining.

We begin by viewing residual RL and diffusion steering as instances of a broader class of *policy-modulation* methods, which adapt pretrained generative imitation-learning policies by modulating their inputs or outputs rather than updating model parameters. Residual policy learning (Ankile et al., 2024b) enables local action refinements but struggles to induce global behavioral changes, whereas diffusion steering (Wagenmaker et al., 2025) allows global modulation but offers limited fine-grained control in off-distribution states. These complementary limitations motivate a unified perspective.Motivated by this perspective, we introduce *Residual Flow Steering (RFS)*, a policy-modulation framework for adapting pretrained generative policies using reinforcement learning. RFS jointly modulates the initial latent noise and applies an affine residual correction to the policy output, enabling global semantic adaptation alongside local dexterous refinement.

We instantiate RFS for multi-finger manipulation, developing controllers that achieve robust multi-object grasping in simulation and transfer effectively to the real world. These controllers can be further improved using limited human demonstrations, enabling data-efficient adaptation to real-world dynamics while preserving precision and dexterity.The contributions of this work are: (1) We introduce *residual flow steering* (RFS), an efficient framework for adapting pretrained flow-matching policies. (2) RFS enables effective simulation pretraining from human demonstrations, capturing complex multi-finger coordination behaviors. (3) RFS supports real-world fine-tuning via offline reinforcement learning, outperforming bootstrapped imitation-learning baselines. (4) We conduct extensive ablations to identify the key components of RFS in both simulation and real-world settings.

## 2 RELATED WORK

**Reinforcement Learning with Demonstrations.** Imitation learning (IL) is commonly used to bootstrap reinforcement learning (RL) for complex robotic tasks by constraining early exploration and providing strong initial behaviors. Prior work incorporates demonstrations via imitation losses, replay buffers, or reset strategies (Nair et al., 2018; Qin et al., 2022; Xiao et al., 2025a; Hu et al., 2024), while other approaches combine demonstration data with online rollouts to improve exploration efficiency (Ball et al., 2023; Hester et al., 2017; Ankile et al., 2025; Song et al., 2023). Demonstration-guided RL has also been applied to high-DoF dexterous manipulation (Rajeswaran et al., 2018b; Haldar et al., 2023; Wang et al., 2024), demonstrating that while demonstrations accelerate learning, they remain insufficient for achieving the fine-grained precision required in challenging dexterous tasks.

**Offline-to-online fine-tuning.** Offline-to-online fine-tuning is widely used to improve sample efficiency in robotic RL: a policy is first pretrained on demonstrations and then refined with limited interaction. Lightweight offline-to-online methods (Kostrikov et al., 2021; Nair et al., 2021; Nakamoto et al., 2024) are popular; however, because they operate directly in action space, they often struggle with distribution shift and the fine-grained corrections required for dexterous control. Recent work applies RL to diffusion policies (Ren et al., 2024b; Fang et al., 2025; Davey and Zheng, 2025) and flow-matching policies (Zhang et al., 2025b; McAllister et al., 2025; Park et al., 2025); however, the iterative refinement structure of generative models and the high dimensionality of action-chunked policies make stable RL fine-tuning particularly challenging and prone to instability.

**Policy Modulation.** Policy modulation methods adapt pretrained policies by modifying either their outputs or their inputs. Output modulation adjusts a base policy's actions using residual policies (Ankile et al., 2024a; Zeng et al., 2020; Silver et al., 2019; Alakuijala et al., 2021; Davchev et al., 2022; Carvalho et al., 2022; Yuan et al., 2024; Xiao et al., 2025b), enabling effective local corrections but unable to induce global behavioral changes when the base policy is limited. Input modulation methods, such as DSRL (Wagenmaker et al., 2025), instead adjust the latent noise of diffusion policies to modify high-level behavior; however, they remain constrained to the demonstration manifold, limiting off-manifold exploration and fine-grained refinement.

## 3 BACKGROUND

### 3.1 FLOW MATCHING

Flow matching (Lipman et al., 2023; Liu, 2022) is a generative modeling framework that learns a time-dependent velocity field that transports a base distribution $p_0$ to a target distribution $p_1$. A

sample evolves according to the ODE $\frac{dx_t}{dt} = v_\theta(x_t, t)$, and the objective is to align the pushforward of $p_0$ with $p_1$ while avoiding ODE backpropagation and likelihood computation by supervising conditional paths rather than full marginals (Lipman et al., 2023). Training draws $(x_0, x_1) \sim p_0 \times p_1$, samples $t \sim \mathcal{U}[0, 1]$, forms the interpolation $x_t = (1-t)x_0 + tx_1$, and regresses the predicted velocity toward the straight-path target $v^\star(x_t, t) = x_1 - x_0$:

$$\mathcal{L}(\theta) = \mathbb{E}_{x_0 \sim p_0, \, x_1 \sim p_1, \, t \sim \mathcal{U}[0,1]} \big\| v_\theta(x_t, t) - v^\star(x_t, t) \big\|^2. \tag{1}$$

Given a trained velocity field $v_\theta$, samples from $p_1$ are obtained via Euler integration (Lipman et al., 2023), yielding a straightforward sampling path:

$$x^{k+1} = x^k + \Delta t_k \, v_\theta(x^k, t_k), \quad x^0 \sim p_0, \quad x^K \approx x_1. \tag{2}$$

The framework becomes conditional by learning $v_\theta(x_t, t, c)$ with an external variable $c$. In our setting, an expert dataset $\mathcal{D} = \{s_i, a_i\}$ is used to train a conditional policy velocity field. The corresponding training objective (Eq. 3) learns a conditional velocity field $v_\theta(a_t, t, s)$:

$$\mathcal{L}(\theta) = \mathbb{E}_{a_0 \sim p_0, \, (s,a) \sim \mathcal{D}, \, t \sim \mathcal{U}[0,1]} \big\| v_\theta(a_t, t, s) - (a - a_0) \big\|^2, \quad a_t = (1-t)a_0 + ta. \tag{3}$$

Integrating the learned field $v_\theta(a_t, t, s)$ (Eq. 2) produces actions from the policy, enabling expressive multimodal behavior while providing natural control points for downstream adaptation. For notational convenience, we use $\pi_\theta$ and $v_\theta$ interchangeably for flow-based policies.

## 3.2 Reinforcement Learning

For adaptation with reinforcement learning (RL), we consider a Markov decision process (MDP) $\mathcal{M} = (\mathcal{S}, \mathcal{A}, \mathcal{T}, r, \gamma)$, where $\mathcal{S}$ and $\mathcal{A}$ denote the state and action spaces, $\mathcal{T}(s' \mid s, a)$ is the transition kernel, $r$ is the reward function, and $\gamma \in [0, 1)$ is the discount factor. The goal is to learn a stochastic policy $\pi_\theta(a \mid s)$ that maximizes the expected discounted return $J(\theta) = \mathbb{E}_{\tau \sim p_\pi}[\sum_{t=0}^\infty \gamma^t r(s_t, a_t)]$. The choice of RL algorithm is not central to our method; instead, we focus on two policy-modulation paradigms that underlie our approach.

**Output modulation.** A central example is *residual reinforcement learning*, in which a small residual policy $\pi_r(a \mid s)$ is trained via RL to correct errors in a pretrained base policy $\pi_\theta(a \mid s)$ (Ankile et al., 2024b; Silver et al., 2019). Action execution is modified via an additive correction, as shown in Eq. 4.

**Input modulation.** A complementary line of work modulates the *inputs* to generative policies. Diffusion steering (Wagenmaker et al., 2025) adapts pretrained diffusion models by learning a policy over the initial latent noise rather than sampling it from a fixed Gaussian. Action generation is performed by integrating the flow-matching dynamics in Eq. 2, which maps an initial latent noise $a_0$ to an action via a denoising process. Diffusion steering optimizes Eq. 5 by adjusting $a_0$ to steer the generative model's behavior through this denoising process.

$$\max_{\pi_r} \mathbb{E}_{\substack{s_0 \sim p_0(s), \, s' \sim p(s'|s,a) \\ a = a_r + a_b, \, a_r \sim \pi_r(a|s), \\ a_b \sim \pi_\theta(a|s)}} \left[ \sum_{t=0}^\infty \gamma^t r(s_t, a_t) \right] \qquad \max_{\pi_{\mathrm{DS}}} \mathbb{E}_{\substack{s_0 \sim p_0(s), \, s' \sim p(s'|s,a) \\ a_0 \sim \pi_{\mathrm{DS}}(a_0|s), \, a \sim p_\theta(a|s,a_0)}} \left[ \sum_{t=0}^\infty \gamma^t r(s_t, a_t) \right]$$
$$\tag{4} \qquad\qquad\qquad\qquad\qquad\qquad\qquad\qquad\qquad\qquad\qquad \tag{5}$$

# 4 Residual Flow Steering: Method and Applications

We propose *Residual Flow Steering* (RFS), a unified framework that combines latent-noise steering for global adaptation with residual actions for precise local corrections on top of a pretrained base policy. This formulation enables efficient learning in high-dexterity tasks and supports robust sim-to-real transfer.

## 4.1 Unified Policy Modulation Framework

### 4.1.1 Policy modulation algorithms for adapting pretrained generative models

Since residual RL and DSRL represent output and input modulation, respectively, it is natural to view them under a unified policy-modulation perspective. To make this connection explicit, we examine the structure of residual RL (Ankile et al., 2024b; Silver et al., 2019) and latent-noise steering (Wagenmaker et al., 2025; Du and Song, 2025) given a pretrained generative policy $v_\theta(a_t, t, s)$ and a denoising-based sampling function $a \sim \text{Des}(s, a_0, v_\theta)$ with $a_0 \sim \mathcal{N}(0, I)$. We place the two objectives side by side:

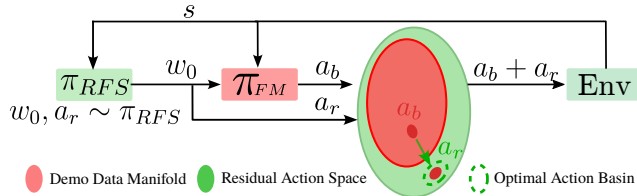

Figure 1: **Residual Flow Steering (RFS).** Given a state $s$, the RFS policy $\pi_{\text{RFS}}$ outputs a latent flow variable $w_0$ and a residual action $a_r$, which jointly steer a pretrained base policy $\pi_{\text{FM}}$ to produce the final action $a_b + a_r$. RFS enables both global mode shifting and fine-grained residual correction, allowing the policy to expand beyond the demonstration data manifold.

$$\max_{\pi_{\text{DS}}} \mathbb{E}_{\substack{s_0 \sim p_0(s) \\ s' \sim p(s'|s,a) \\ a = \text{Des}(s, a_0, v_\theta), \\ a_0 \sim \pi_{\text{DS}}(a_0|s)}} \left[ \sum_{t=0}^{\infty} \gamma^t r(s_t, a_t) \right] \qquad (6)$$

$$\max_{\pi_r} \mathbb{E}_{\substack{s_0 \sim p_0(s) \\ s' \sim p(s'|s,a) \\ a = a_r + a_b, \; a_r \sim \pi_r(a|s) \\ a_b = \text{Des}(s, a_0, v_\theta), \\ a_0 \sim \mathcal{N}(0, I)}} \left[ \sum_{t=0}^{\infty} \gamma^t r(s_t, a_t) \right] \qquad (7)$$

Both objectives modulate the behavior of the base policy $v_\theta$—either by adjusting the latent noise or applying an affine action correction. More generally, policy modulation for a generative policy $v_\theta(a_t, t, s)$ optimizes parametric input ($g$) or output ($f$) transformations without modifying the underlying parameters $\theta$ as formalized in Eq. 8.

### 4.1.2 RESIDUAL FLOW STEERING FOR JOINT LOCAL AND GLOBAL ADAPTATION

We instantiate the modulation functions $f$ and $g$ as *Residual Flow Steering* (RFS), which unifies residual RL for **output modulation** (Silver et al., 2019; Zhang et al., 2020) with diffusion-steering RL for **input modulation** (Wagenmaker et al., 2025). Input modulation adjusts the latent noise of a flow-matching policy for *global* behavioral variation via $a_0 = \pi_H(a_0 \mid s)$, while output modulation applies *local* refinements through the affine correction $a = \text{Des}(s, a_0, v_\theta) + a_r$ with $a_r \sim \pi_r(a \mid s)$. RFS combines these mechanisms through a unified modulation policy $\pi_{\text{RFS}}(a_0, a_r \mid s)$, jointly producing $a_0$ for global steering and $a_r$ for precise local adjustments. Here, $a_0$ shapes the generative model's overall behavior, while $a_r$ compensates for off-manifold requirements or imperfections in the base policy $v_\theta$. The resulting objective in Eq. 8 is compatible with standard RL algorithms, and the next subsection instantiates RFS for dexterous manipulation in simulation and the real world.

$$\max_{\phi} \mathbb{E}_{\substack{s_0 \sim p_0(s), \; s' \sim p(s'|s,a) \\ a = f_\phi(a_b, s), \; a_0 = g_\phi(s) \\ a_b \sim \text{Des}(s, a_0, v_\theta)}} \left[ \sum_{t=0}^{\infty} \gamma^t r(s_t, a_t) \right] \qquad (8)$$

$$\max_{\pi_{\text{RFS}}} \mathbb{E}_{\substack{s_0 \sim p_0(s), \; s' \sim p(s'|s,a) \\ a_0, a_r \sim \pi_{\text{RFS}}(a_0, a_r|s) \\ a_b \sim \text{Des}(s, a_0, v_\theta) \\ a = a_r + a_b}} \left[ \sum_{t=0}^{\infty} \gamma^t r(s_t, a_t) \right] \qquad (9)$$

### 4.2 INSTANTIATING RFS FOR DEXTEROUS MANIPULATION

We instantiate RFS for dexterous manipulation through a simulation-to-reality pipeline that enables online fine-tuning in simulation and offline fine-tuning using limited real-world data, applied to state- and vision-based flow-matching policies, respectively, enabling broad exploration and data-efficient adaptation (Fig. 2). We demonstrate the approach across several dexterous manipulation tasks.

### 4.2.1 SIMULATION TRAINING: DATA GENERATION AND ONLINE RL

Dexterous manipulation poses a key challenge in effective exploration over high-dimensional action spaces. We collect a small set of VR-teleoperated demonstrations **entirely in simulation** to train a base generative policy $v_\theta$ using a flow-matching objective (Lipman et al., 2022). Although imperfect

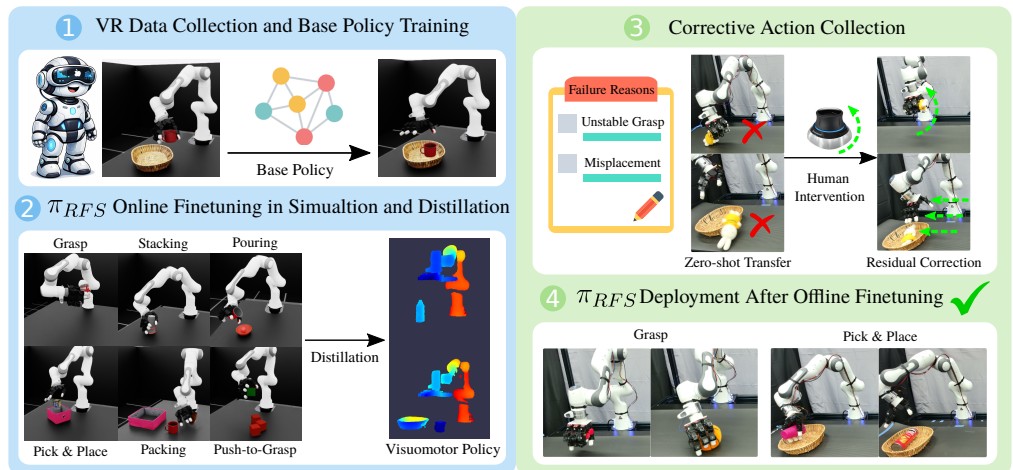

Figure 2: **Overview of the sim-to-real Residual Flow Steering (RFS) pipeline.** (1) VR teleoperation is used to collect demonstrations across multiple manipulation tasks to train task-specific flow-matching base policies. (2) In simulation, the RFS policy $\pi_{\text{RFS}}$ is fine-tuned on top of each base policy and distilled into task-specific visuomotor policies to improve sim-to-real transfer. (3) During zero-shot real-world deployment, human corrective actions correct execution failures such as unstable grasps and misplacement. (4) These corrected transitions are used for offline fine-tuning of $\pi_{\text{RFS}}$ on a Franka–Leap Hand system, improving real-world grasping and pick-and-place performance.

in isolation, this policy captures coordinated hand–arm behaviors and provides a strong initialization for reinforcement learning. We then train the RFS policy $\pi_{\text{RFS}}$ with **online PPO** to optimize grasp success and stability.

The trained RFS policy is used to generate additional simulation demonstrations with access to privileged low-level state information $s$ (e.g., object poses and velocities), which are distilled into point-cloud–conditioned visuomotor policies via a standard student–teacher framework (Chen et al., 2021).

### 4.2.2 REAL-WORLD FINE-TUNING: OFFLINE RFS

Direct transfer of the distilled visuomotor policy $v_\phi(a_t, o_{\text{pc}}, s_{\text{pro}}, t)$ to the real world often fails under novel objects or varied initial conditions. To bridge this gap, we apply an *offline* RFS fine-tuning stage using a small human-corrected dataset. During data collection, the base policy executes rollouts while a human provides corrective actions at visited states, yielding $\mathcal{D} = \{((o, s), (a_{\text{human}}, a_b, a_0), (o', s'), r)\}$. Residual actions are defined as $a_r = a_{\text{human}} - a_b$, producing the RFS-ready dataset $\mathcal{D}_{\text{RFS}} = \{((o, s), (a_0, a_r), (o', s'), r)\}$. We then train $\pi_{\text{RFS}}(a_0, a_r \mid o_{\text{pc}}, s_{\text{pro}})$ with offline reinforcement learning (Levine et al., 2020), using TD3+BC (Fujimoto and Gu, 2021) for actor–critic updates with imitation regularization.

**Critic Update.** Given $\mathcal{D}_{\text{RFS}}$, we apply a standard TD-learning critic update in Eq. 10. The critic is conditioned on the combined actions $a$ and $a'$ rather than the decomposed components $(a_0, a_r)$; Section 5.2 provides an empirical justification for this design choice.

$$\min_\phi \mathop{\mathbb{E}}_{\substack{((o,s),(a_0,a_r),(o',s'),r)\sim\mathcal{D}_{\text{RFS}} \\ a_0',a_r'\sim\pi_{\text{RFS}}(\cdot|o',s') \\ a_b\sim\text{Push}(o,s,a_0,v_\phi),\ a=a_b+a_r \\ a_b'\sim\text{Push}(o',s',a_0',v_\phi),\ a'=a_b'+a_r'}} \left[\|Q_\phi(o,s,a) - r - \gamma Q_{\bar\phi}(o',s',a')\|^2\right].$$

(10)

**Actor Update.** Following TD3+BC (Fujimoto and Gu, 2021), the actor maximizes the critic while applying a BC regularizer on the residual actions:

$$\arg\max_{\pi_{\text{RFS}}} \mathop{\mathbb{E}}_{\substack{((o,s),(a_0,a_r),(o',s'),r)\sim\mathcal{D}_{\text{RFS}} \\ \hat a_0,\hat a_r\sim\pi_{\text{RFS}}(\cdot|o,s) \\ \hat a_b\sim\text{Push}(o,s,\hat a_0,v_\phi),\ \hat a=\hat a_b+\hat a_r}} \left[Q(o,s,\hat a) - \lambda_{\text{BC}} \|\hat a_r - a_r\|^2\right].$$

(11)

## 5 EXPERIMENTS

We structure our experiments around the following research questions: **Q1:** Does residual refinement improve success in dexterous manipulation tasks in simulation? **Q2:** Which design choices are most effective across both simulation and real-world settings? **Q3:** How does latent-noise modulation influence real-world adaptation performance? As outlined in Section 4.2, we first evaluate RFS for simulation data generation and then for real-world fine-tuning. Our experiments focus primarily on dexterous grasping with a multifingered hand, though the method generalizes to broader manipulation tasks.

### 5.1 RFS FOR SIMULATION DATA GENERATION AND ONLINE RL TRAINING

#### 5.1.1 SETUP

We evaluate RFS on six simulated tasks: grasping , pick-and-place, non-prehensile push-to-grasp, long-horizon packing, and high-precision stacking and pouring (Fig. 4). Using Apple Vision Pro AR, we collect approximately 400 demonstrations per task and pretrain a generative base policy $v_\theta(a_t, t, s)$ using flow matching (Lipman et al., 2022). Base-policy success rates are reported in Table 1. Although the pretrained policy achieves only moderate success, it provides strong motion priors that RFS further refines via PPO (Schulman et al., 2017b) (Sec. 4.2.1).

Simulation Objects     Real Objects

Figure 3: Simulation and real objects used for dexterous grasping and pick & place.

#### 5.1.2 BASELINES AND ABLATIONS

**Baselines.** We compare RFS against a broad suite of reinforcement-learning baselines commonly used for simulation-driven dexterous manipulation. All demonstration-based baselines use the same offline data as RFS. We group these methods into four categories based on their underlying adaptation strategies: **(1) Diffusion/Flow RL Fine-tuning.** DPPO (Ren et al., 2024a) and ReinFlow (Zhang et al., 2025a), which fine-tune demonstration-pretrained diffusion or flow-matching policies through reinforcement learning with environment interaction. **(2) Offline-to-Online RL.** IQL (Kostrikov et al., 2021), AWAC (Nair et al., 2021), and Flow Q-Learning (Park et al., 2025), which are initialized from offline demonstrations and subsequently improved via online interaction. **(3) RL with Demonstrations.** RLPD (Ball et al., 2023) and IBRL (Hu et al., 2024), which incorporate demonstration data directly into policy updates.

**Ablations.** To isolate the contributions of *both* residual refinement and latent-noise steering, we conduct targeted ablations. (1) **DSRL (Wagenmaker et al., 2025):** steering the latent noise of the base policy *without* residual actions. (2) **State-of-the-Art Residual RL:** strong residual-learning baselines, including Policy Decorator (Yuan et al., 2024) and ResiP (Ankile et al., 2024a).

#### 5.1.3 EMPIRICAL RESULTS IN SIMULATION

**Comparison to Baselines:** We summarize the performance of all baselines in Table 1 and analyze their behavior as follows:

**(1) Diffusion-/Flow-Based RL Fine-Tuning.** DPPO (Ren et al., 2024a) underperforms across all tasks, with all success rates below 0.50. ReinFlow Zhang et al. (2025a) shows modest improvements (0.58 grasp, 0.46 pick & place, 0.39 packing), but fails on push-to-grasp, stacking, and pouring. In contrast, restricting RL updates to the initial noise and residual terms (RFS) yields more stable learning, achieving success rates at least 0.35 higher across all tasks than applying RL over the full denoising trajectory.

**(2) Offline-to-Online RL.** IQL (Kostrikov et al., 2021) is the strongest offline-to-online baseline (0.69 grasp, 0.56 pick-and-place, 0.63 stack, 0.59 pour), but struggles on packing (0.20) and push-to-grasp (0.26). AWAC (Nair et al., 2021) and FlowQ (Park et al., 2025) collapse on multiple tasks.

Overall, offline-to-online RL methods lack directed exploration, whereas RFS leverages base-policy pretraining to achieve consistently higher success rates across all tasks.

(3) **RL with Demonstrations.** RLPD (Ball et al., 2023) performs well on grasping (0.54) and pick-and-place (0.76), but collapses on tasks requiring long-horizon reasoning (e.g., packing) or high-precision manipulation (e.g., stacking and pouring). IBRL (Hu et al., 2024) succeeds only on grasping, packing, and pouring. In contrast, RFS consistently improves over the base policy across all tasks.

**Comparison to Ablations.** We additionally evaluate strong ablations and related adaptation strategies (Table 1).

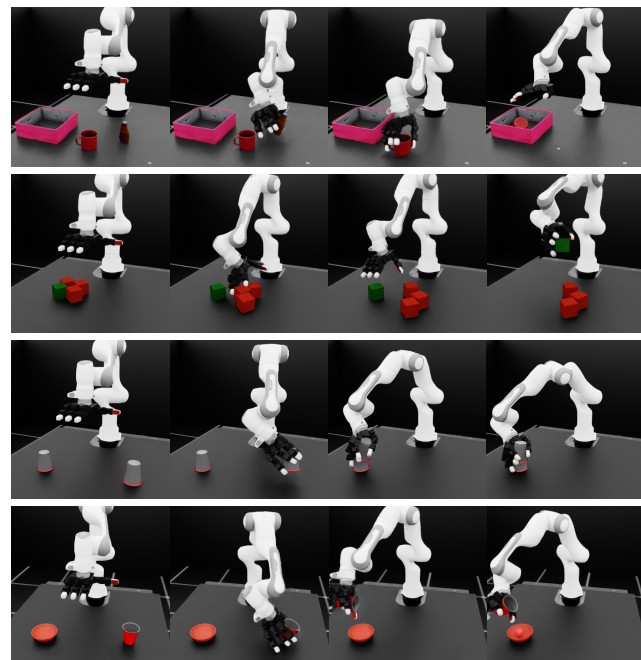

**(1) Residual RL methods.** Policy Decorator (Yuan et al., 2024) and ResiP (Ankile et al., 2024a) outperform earlier baselines, but their gains remain largely local, achieving moderate success on grasping ($\sim$ 0.60) and pick-and-place ($\sim$ 0.50). Performance degrades on sequential and high-precision tasks, including packing (0.40), push-to-grasp (0.15), stacking (0.30), and pouring (0.20). In contrast, RFS achieves $\sim$ 0.80 success on packing and push-to-grasp and $\sim$ 0.90 on the remaining tasks.

**(2) DSRL (Wagenmaker et al., 2025).** DSRL is the strongest baseline, achieving 0.74 (grasp), 0.70 (pick-and-place), 0.64 (packing), and 0.43 (push-to-grasp), but only 0.15 on stacking and 0.26 on pouring. Because its corrections are constrained to the demonstration distribution, DSRL struggles to adapt to challenging or out-of-distribution states.

Figure 4: Representative rollouts for the dexterous manipulation tasks. From top to bottom: Packing, Push-to-Grasp, Packing and Stacking.

**Overall** By jointly modulating the input and output of the base policy, RFS preserves base-policy performance while consistently improving upon it, achieving the highest success rates across all tasks: 0.89 (Grasping), 0.94 (Pick & Place), 0.78 (Packing), 0.72 (Push-to-Grasp), 0.95 (Stacking), and 0.87 (Pouring), with an average success rate of 0.87.

| Category | Method | Grasping | Pick & Place | Packing | Push-to-Grasp | Stacking | Pouring | Avg |
|---|---|---|---|---|---|---|---|---|
| **Base Policy** | Flow Matching (Lipman et al., 2022) | 0.495 | 0.367 | 0.30 | 0.131 | 0.06 | 0.15 | 0.250 |
| **Diffusion / Flow RL Finetuning** | DPPO (Ren et al., 2024a) | 0.41 ± 0.05 | 0.433 ± 0.102 | 0.186 ± 0.023 | 0.04 ± 0.036 | 0.00 ± 0.00 | 0.00 ± 0.00 | 0.178 ± 0.035 |
| | ReinFlow (Zhang et al., 2025a) | 0.584 ± 0.043 | 0.462 ± 0.081 | 0.100 ± 0.089 | 0.398 ± 0.236 | 0.59 ± 0.021 | 0.32 ± 0.15 | 0.409 ± 0.103 |
| **Offline-to-Online RL** | IQL (Kostrikov et al., 2021) | 0.690 ± 0.034 | 0.560 ± 0.075 | 0.203 ± 0.063 | 0.267 ± 0.108 | 0.625 ± 0.073 | 0.583 ± 0.203 | 0.488 ± 0.093 |
| | AWAC (Nair et al., 2021) | 0.485 ± 0.163 | 0.240 ± 0.120 | 0.00 ± 0.00 | 0.00 ± 0.00 | 0.780 ± 0.14 | 0.625 ± 0.08 | 0.355 ± 0.066 |
| | Flow Q-Learning (Park et al., 2025) | 0.050 ± 0.012 | 0.486 ± 0.110 | 0.00 ± 0.00 | 0.381 ± 0.040 | 0.00 ± 0.00 | 0.00 ± 0.00 | 0.153 ± 0.027 |
| **RL with Demonstrations** | RLPD (Ball et al., 2023) | 0.547 ± 0.056 | 0.763 ± 0.059 | 0.00 ± 0.00 | 0.012 ± 0.024 | 0.053 ± 0.13 | 0.68 ± 0.13 | 0.343 ± 0.067 |
| | IBRL (Hu et al., 2024) | 0.404 ± 0.216 | 0.00 ± 0.00 | 0.432 ± 0.150 | 0.00 ± 0.00 | 0.00 ± 0.00 | 0.36 ± 0.10 | 0.199 ± 0.078 |
| **Residual RL (State-of-the-Art)** | Policy Decorator (Yuan et al., 2024) | 0.52 ± 0.09 | 0.314 ± 0.160 | 0.535 ± 0.021 | 0.176 ± 0.016 | 0.00 ± 0.00 | 0.17 ± 0.23 | 0.286 ± 0.086 |
| | ResiP (Ankile et al., 2024a) | 0.65 ± 0.096 | 0.57 ± 0.14 | 0.302 ± 0.028 | 0.165 ± 0.18 | 0.67 ± 0.05 | 0.24 ± 0.12 | 0.433 ± 0.081 |
| **DSRL (Wagenmaker et al., 2025)** | | 0.732 ± 0.04 | 0.692 ± 0.012 | 0.639 ± 0.024 | 0.430 ± 0.085 | 0.135 ± 0.015 | 0.268 ± 0.010 | 0.483 ± 0.031 |
| **RFS (Ours)** | | **0.899 ± 0.026** | **0.939 ± 0.012** | **0.781 ± 0.019** | **0.721 ± 0.022** | **0.951 ± 0.006** | **0.873 ± 0.010** | **0.861 ± 0.016** |

Table 1: Success rates across tasks and RL methods.

## 5.2 REAL-WORLD OFFLINE FINETUNING

Using data generated by RFS in simulation with access to privileged state information, we distill a point cloud–based policy via a standard student–teacher framework (Chen et al., 2021) for real-world deployment.

We then evaluate offline RFS for adapting the distilled policy in real-world grasping and pick-and-place tasks with both seen and unseen objects (Fig. 1, Fig. 3). Experiments are conducted on a Franka arm equipped with a LEAP hand under Cartesian impedance control at 10 Hz. We evaluate seven objects, including two shared with simulation and five novel deformable objects. Compared to rigid simulation assets, real-world objects exhibit varying compliance, contact dynamics, and appearance, introducing a sim-to-real gap that often causes pretrained policies to fail.

For offline RL fine-tuning, we collect 50 human corrective demonstrations using a SpaceMouse (Appendix A.5.1). Rewards are defined programmatically by segmenting and tracking objects with SAM2 (Ravi et al., 2024) and extracting their centroids (Appendix A.5.2).

### 5.2.1 BASELINES AND ABLATIONS

**Baselines.** We evaluate RFS with offline RL (Section 4.2.2) as a data-efficient approach for fine-tuning simulation-trained policies. We compare against three adaptation strategies: (1) **Zero-Shot Transfer**, which deploys the distilled simulation policy without fine-tuning; (2) **BC Fine-tuning**, which uses 50 real-world demonstrations to further train the distilled policy via flow matching; (3) **Co-Training**, which fine-tunes the distilled policy by mixing 50 real-world demonstrations with additional simulated rollouts under the flow-matching objective. These comparisons demonstrate that offline RL fine-tuning yields substantially stronger adaptation than standard flow-matching–based updates.

**Critic Design Choices:** We additionally study alternative critic architectures in the offline-RL setting, comparing: (1) $Q(a_r, o)$, which conditions only on residual actions and observations; (2) $Q([a_r, a_b], o)$, which conditions on both residual and base-policy actions; (3) $Q(a_b + a_r, o)$ **(Ours)**, which conditions solely on the final executed action. This isolates the effect of critic input structure on offline-RL stability and performance.

**Ablations:** We further analyze offline RL using only residual actions (Residual RL) and only latent-noise steering (DSRL) to quantify their individual contributions.

### 5.3 EMPIRICAL RESULTS IN THE REAL WORLD

**Comparison to Supervised Fine-Tuning.** As shown in Fig. 5 and Fig. 6, the zero-shot policy performs poorly, exhibiting translational and rotational offsets, insufficient finger closure during grasping, and mistimed or misplaced actions in pick-and-place, highlighting a significant sim-to-real gap. While co-training substantially improves performance on known objects (83.3% grasping and 60% pick-and-place success), generalization to unseen objects remains limited. The noisy and inconsistent nature of human-collected data introduces instability and rollout failures, underscoring the difficulty of generalizing to novel objects.

**Comparison to Variants of Offline RL.** As discussed in Sec. 4.2.2, TD3+BC conditions the critic on the actor's output, allowing different Q-function parameterizations. Conditioning the critic on either $Q(a_r, o)$ or $Q([a_r, a_b], o)$ fails to produce consistent grasp poses or precise placement, yielding zero success on both grasping and pick-and-place. In contrast, conditioning on the executed action $Q(a_b + a_r, o)$—used by both Residual RL and

| Method | Pick-and-Place | Grasping |
|---|---|---|
| 0-shot | $50.0 \pm 0.0$ | $43.3 \pm 6.2$ |
| Co-training | $60.0 \pm 0.0$ | $83.3 \pm 0.0$ |
| BC | $40.0 \pm 0.0$ | $73.3 \pm 0.0$ |
| Residual RL | $50.0 \pm 0.0$ | $80.0 \pm 0.0$ |
| DSRL | $80.0 \pm 0.0$ | $70.0 \pm 0.3$ |
| **RFS (Ours)** | $\mathbf{90.0 \pm 0.0}$ | $\mathbf{80.0 \pm 2.9}$ |

Table 2: Performance on **seen objects**. Values report mean success rate with **95% confidence interval** across trials.

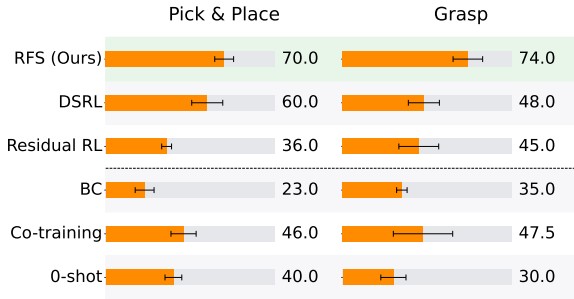

Figure 5: Real-world evaluation results on **unseen objects**. Bars show mean success rates for **Pick-and-Place** (left) and **Grasping** (right), with horizontal error bars denoting the **95% confidence interval**. Numeric annotations report the mean success rate (±95% CI).

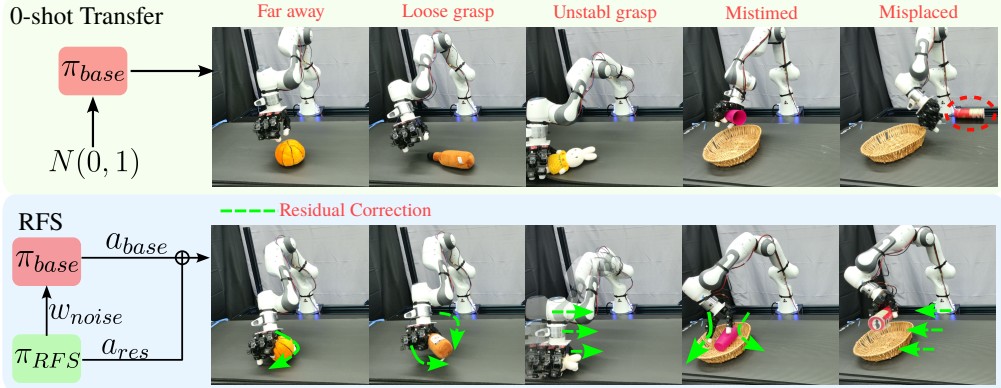

Figure 6: **Common failure modes in real-world manipulation and RFS corrections.** We illustrate five representative failure cases of zero-shot sim-to-real transfer, including early hand closure, loose or unstable grasp poses, and mistimed or misplaced actions during pick-and-place. The top row shows failures produced by the zero-shot policy, while the bottom row shows the corrected outcomes after applying *Residual Flow Steering (RFS)*. By jointly correcting action timing, grasp pose, and target location, RFS enables more stable and successful task execution.

RFS (Fig. 5)—achieves substantially higher performance across all baselines. These results indicate that directly coupling residual actions with the executed base actions is critical for stable and effective offline RL adaptation.

**Comparison to Ablations of RFS.** We evaluate RFS variants that use only residual actions ($\pi(a_r \mid o)$) or only latent-noise steering via DSRL ($\pi(a_0 \mid o)$). As shown in Fig. 5 and Fig. 6, the full model $\pi(a_0, a_r \mid o)$ achieves the strongest performance, particularly on unseen objects in both grasping and pick-and-place tasks. Qualitatively, latent-noise steering facilitates global exploration, while residual actions enable precise local refinement, making their combination more effective than either component alone.

**Comparison to Direct Real-Data Fine-Tuning.** Fine-tuning RFS on a policy trained solely on real demonstrations yields only modest gains ($70\% \to 73\%$ on seen objects and $35\% \to 50\%$ on unseen objects for grasping, with little improvement for pick-and-place), reflecting the limited coverage of the real-data distribution. In contrast, simulation pretraining provides broad coverage over grasps, poses, and perturbations, enabling adaptation behaviors that cannot be learned from real data alone. As a result, fine-tuning a simulation-pretrained policy with a small number of real demonstrations yields substantially larger improvements and stronger generalization.

## 6 CONCLUSION

We introduced Residual Flow Steering (RFS), a reinforcement learning framework that unifies latent noise steering for global exploration with residual corrections for local refinement. Across both simulation and real-world dexterous manipulation tasks, RFS consistently outperforms conventional baselines—including residual-only methods and diffusion-based steering—by enabling effective simulation pretraining and data-efficient fine-tuning in the real world. Despite these strong results, several limitations remain and point to directions for future work. First, our current implementation primarily relies on point cloud observations, which provide limited semantic and contextual understanding. As a result, performance can degrade in cluttered or semantically rich environments that require higher-level reasoning. Second, real-world adaptation in our framework is currently restricted to offline fine-tuning. This prevents the policy from updating online in response to dynamically changing conditions, limiting its responsiveness and robustness during real-world deployment.

ACKNOWLEDGMENTS

This work was supported by Amazon FAR (Frontier AI & Robotics).

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

# A APPENDIX

## A.1 THE USE OF LARGE LANGUAGE MODELS

In this paper, we primarily use LLMs for language polishing, making our writing more concise and accessible. We provide raw drafts and ask the LLM to refine them. We also use LLMs for guidance on figure preparation, such as creating illustrations in Inkscape and adjusting figure size and font in Matplotlib.During rebuttal, we also used an LLM to polish the writing, making the language more concise and clear based on reviewer feedback.

## A.2 SIMULATION SETUP AND TRAINING

### A.2.1 REINFORCEMENT LEARNING DESIGN CHOICES

To enhance robustness and mitigate controller misalignment for sim-to-real transfer, we introduce the following design choices:

**Simulator:** All of our simulated tasks are built on IsaacLab (NVIDIA et al., 2025).

**Teleoperation:** We further developed the teleoperation system in IsaacLab (NVIDIA et al., 2025) to make it more suitable for dexterous hand manipulation.

**External Disturbances:** Apply random external disturbances to each joint at intervals of 2–5 steps, encouraging the policy to recover from perturbations.

**Observation Space:** In the simulation RL training, we use state-based information, including the robot proprioception, the current object pose, the target object pose, and binary contact signals between each fingertip and the object.

**Data Collection:** We use the Vision Pro for data collection, leveraging an application built on IsaacLab Mittal et al. (2023).

## A.3 PPO HYPERPARAMETERS

We train the Residual Flow Steering (RFS) policy using Proximal Policy Optimization (PPO) (Schulman et al., 2017a) during online fine-tuning in simulation. The same set of hyperparameters is used across all tasks.

**Policy and Value Networks.** Both the RFS policy $\pi_{\text{RFS}}(a_0, a_r \mid s)$ and the value function $V(s)$ are parameterized by multilayer perceptrons with three hidden layers of width 256.128,64 and ReLU activations. The policy outputs both the latent noise steering variable $w_0$ for flow matching base policy and the residual action $a_r$.

Table 3: PPO hyperparameters used for online RFS training in simulation.

| Hyperparameter | Value |
|---|---|
| Discount factor $\gamma$ | 0.99 |
| GAE parameter $\lambda$ | 0.95 |
| Policy learning rate | $3 \times 10^{-4}$ |
| Value learning rate | $1 \times 10^{-3}$ |
| Optimizer | Adam |
| Clip range $\epsilon$ | 0.2 |
| Value loss coefficient | 0.5 |
| Max gradient norm | 1.0 |
| Minibatch size | 1024 |

### A.3.1 POLICY DISTILLATION

To improve sim-to-real transfer, we use point clouds as visual observations. To mitigate errors from camera calibration and hardware variations, we collect simulation data from multiple camera

viewpoints and calibrate all point clouds to the robot base frame. We further inject random noise into the camera extrinsics during data collection and into the point clouds during training to improve robustness. In simulation, we collect 1,000 demonstrations for each of the grasping and pick-and-place tasks to train the policy.

### A.4 REAL WORLD EXPERIMENT DETAILS

#### A.4.1 REAL ROBOT EVALUATION

For real-robot evaluation, we tested our policy within a $30 \times 50$ cm region located 0.35–0.65 m from the robot base and spanning $-0.25$ to $0.25$ m horizontally. We evaluated known objects with 20 trials each, and seven unknown objects with 10 trials each.

### A.5 TELEOPERATION DATA COLLECTION

For real-world teleoperation data collection in the co-training setup, we developed a Vision Pro application based on Park and Agrawal (2024) that supports teleoperation at a controlled frequency of 10Hz.

#### A.5.1 OFFLINE DATA COLLECTION

When the hand is within $\sim 10$ cm of the table—where most failures occur (Fig. 6), we enable human intervention via a SpaceMouse (Fig. 8). Rather than granting full manual control, which would shift the policy distribution, we compute residual corrections as bounded deltas from the base policy output, conditioned on the current observation. Specifically, the CFM module predicts the next action, and we take the difference between this rollout and the operator's input. Residuals are constrained to $\leq 1.5$ cm in Cartesian translation and $\leq 0.05$ rad for finger motion, applied uniformly across all joints. In practice, corrections were limited to Cartesian translation and minor finger adjustments, with the SpaceMouse $z$-axis mapped to finger motion.

#### A.5.2 REWARD FUNCTION DESIGN

In the real world, the reward is constructed using SAM2 (Ravi et al., 2024) to track the object in image space and extract its point cloud, from which the object center is computed. The reward consists of two terms: (1) the distance between the object center and the palm, obtained via forward kinematics, and (2) the object's lifting height, which encourages stable grasp execution. Figure 7 illustrates the reward trend.

### A.6 TD3+BC HYPERPARAMETERS

We fine-tune the Residual Flow Steering (RFS) policy in the real world using the TD3+BC algorithm (Fujimoto and Gu, 2021), operating in an offline setting with a fixed dataset of human-corrected transitions. Observations are represented as point clouds, and the perception backbone is adapted from PointNet (Qi et al., 2017). The TD3+BC hyperparameters used in all real-world experiments are summarized in Table 4.

Table 4: TD3+BC hyperparameters used for offline RFS fine-tuning in real-world experiments.

| Hyperparameter | Value |
|---|---|
| Actor learning rate | $3 \times 10^{-4}$ |
| Critic learning rate | $3 \times 10^{-4}$ |
| Policy update delay | 10 |
| Target policy noise std | 0.2 |
| Target noise clip | 0.5 |
| Behavior cloning weight $\lambda_{BC}$ | 0.2 |
| Batch size | 512 |
| Replay buffer | Fixed offline dataset |

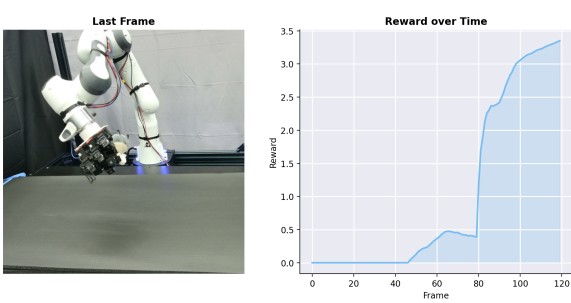

Figure 7: Reward progression over time in real-world experiments.

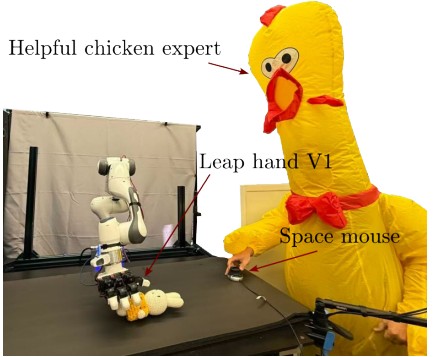

Figure 8: Offline real-world data collection setup using the Leap Hand V1 and a space mouse interface.

## B  SIMULATION BASELINES AND CODEBASES

We compare RFS against a diverse set of simulation baselines spanning diffusion/flow RL fine-tuning, offline-to-online RL, RL with demonstrations, and the SOTA residual RL methods. For each baseline, we use the open-source implementations and follow their recommended training protocols.

- **DPPO** (Ren et al., 2024a):
  `https://github.com/irom-princeton/dppo`
- **ReinFlow** (Zhang et al., 2025a):
  `https://github.com/ReinFlow`
- **Flow Q-Learning (FQL) and IQL** (Park et al., 2025):
  `https://github.com/seohongpark/fql/tree/master/agents`
- **AWAC** (Nair et al., 2021):
  `https://github.com/ikostrikov/jaxrl`
- **RLPD and IBRL** (Hu et al., 2024):
  `https://github.com/hengyuan-hu/ibrl`
- **Policy Decorator** (Yuan et al., 2024):
  `https://github.com/tongzhoumu/policy_decorator`
- **ReSIP** (Ankile et al., 2024a):
  `https://github.com/ankile/robust-rearrangement`

All baselines are trained using the same simulator, observation modalities, and task specifications as RFS to ensure a fair comparison.

