# OpenReview forum: "RFS: Reinforcement learning with Residual flow steering for dexterous manipulation"
_ICLR.cc/2026/Conference — ICLR 2026 Poster_

### Official Review · Reviewer_kxpu · 2025-10-16

**Soundness:** 3
**Presentation:** 2
**Contribution:** 3
**Rating:** 4
**Confidence:** 4

**Summary:**

This paper introduces Residual Flow Steering (RFS), a novel reinforcement learning framework for fine-tuning flow-matching policies in dexterous manipulation.
RFS adapts a pre-trained policy by learning to modulate both the initial latent noise (for global exploration) and output a residual action (for local refinement).
The method is evaluated in both simulation and real-world settings, demonstrating improved performance over several baselines. The core idea is intuitive and addresses a relevant challenge in policy adaptation. However, the paper could be strengthened by a more comprehensive related work section, clearer methodological explanations, and more competitive baseline comparisons.

**Strengths:**

1. This paper proposes a novel RL method for adapting a policy pre-trained with flow-matching to dexterous manipuation tasks.
The adaptation is achieved by training a policy to output an initial noise and a residual action, which correspond to the input and output of the flow policy.
2. The overall ideas and intuition of the method are clear.
3. The paper demonstrate the performance of the proposed method with both simulation and real-world experiments.

**Weaknesses:**

1. A more comprehensive related work section will enhance the paper.
For example, recent papers on residual RL are not included, such as policy decorator[1].
And some baselines compared in this paper are not introduced in the related work.
2. Some details about the method are not clearly explained. Please check the questions in the following section.
3. The chosen baselines, while reasonable, do not fully demonstrate the advantage of RFS over the state-of-the-art.
A more compelling comparison would be against: (1) Recent RL methods specifically designed for fine-tuning diffusion/flow policies (beyond the ablated DSRL) (2) State-of-the-art residual RL methods that finetune a base policy, to better isolate the contribution of the combined approach.

Minor:
1. The title "Imitation-Bootstrapped Reinforcement Learning" in related work might be slightly misleading. As defined in works like IBRL[2], the term has a specific meaning. The papers [3][4] cited under this heading are more broadly categorized as "RL with Demonstration". It would be helpful to refine this terminology for precision.

[1] Zhiyuan Yuan, et al. "Policy Decorator: Model-Agnostic Online Refinement for Large Policy Model." ICLR 2025.

[2] Hengyuan Hu, et al. "Imitation Bootstrapped Reinforcement Learning." RSS 2024.

[3] Ashvin Nair, et al. "Overcoming exploration in reinforcement learning with demonstrations." ICRA 2018.

[4] Aravind Rajeswaran, et al. "Learning complex dexterous manipulation with deep reinforcement learning and demonstrations." RSS 2018.

**Questions:**

1. In section 5.2, it is mentioned that base policy actions $a_b$ and human corrections $a$ are recorded and transformations are applied to them to obtain $(a_0, a_r)$ for training. Why is the initial latent noise $a_0$ not recorded directly?
2. The RFS policy $\pi_{RFS}$ outputs $a_0$, which is passed through the "Push" function (an ODE solver involving multiple evaluations of $v_\theta$). Could you clarify if gradients are backpropagated through the entire "Push" operation?
If so, there will be a huge computational cost for the gradient calculation during training and it is unstable. If not, how are the gradients for $a_0$ obtained?
By the way, could you mention the number of steps used for the "Push" operation?
3. The policy $\pi_{RFS}$ is trained to output both $a_0$ and $a_r$. I am curious about the results if we separate the current loss into two terms. For example, detaching $a_r$ when calculating gradients for the $a_0$, or vice-versa, to see if it stabilizes training or improves performance?
4. The process of using RFS for simulation data generation could be explained more clearly.
I got the following questions when reading this part:
First, are VR-teleoperated demonstrations real-world data or simulation data?
Second, the high-level policy $\pi_{RFS}$ is trained with the same demonstrations using an offline RL method or trained in simulation using online RL method?
I can find the answer in Section 6.1: simulation data, online RL method (PPO) while it will be easier for the reader to understand if the Section 5.1 can be rephrased.
5. What is the specific advantage of applying RFS to create a simulation policy first, rather than directly applying RFS to the original base policy $v_{\theta}(a_t, s, t)$ using human-collected real-world data?

---

> ### Author Response · Authors · 2025-11-25
> **Official Comment by Authors**
>
> Dear Reviewer,
>
> Thank you for your feedback on how we can improve our paper. We respond to individual comments below:
>
> ---
>
> **Q: Baselines of separating the loss into two terms.**
> >Thank you for the suggestion. To ensure we fully understand your suggestion, could you explain what you mean by “separating the loss into two terms” and “detaching one branch when computing gradients for the other”? Are you suggesting a multi-head PPO setup in which the latent-noise and residual-action outputs are optimized with separate gradient paths? Any clarification would help us implement the idea more accurately.
>
> ---
>
> **Q: Baseline Coverage and Related Work Completeness**
> >We expanded our evaluation to include diffusion/flow-policy finetuning (DPPO, ReinFlow), state-of-the-art residual RL (Policy Decorator, ResiP), as shown in Table 1. Performance across these methods remains limited: diffusion/flow finetuning reaches **41–58%**, residual RL **52–74%**, reflecting their reliance on the pretrained sampler or the demonstration manifold and their inability to adapt global behavior. For completeness, we also report offline-to-online RL, including AWAC, IQL, Flow Q-Learning (**22–55%**), and RL with demonstrations, including RLPD, IBRL (**40–55%**), which likewise cannot achieve broad behavioral adaptation. In contrast, RFS reaches 97%, showing that existing methods are restricted to either demonstration-anchored policies or local corrections, whereas RFS provides the global adaptation required for high performance.
> We also updated the related work section to include the recent residual-RL methods and newly added baselines, providing a more straightforward overview of how diffusion/flow finetuning, residual RL, and offline-to-online RL relate to RFS.
>
> ---
>
> **Q: Applying RFS to Real-Only Demonstration Policies**
> >As suggested, we have added an experiment applying RFS directly to a policy trained only on real-world demonstrations, rather than sim-to-real. The improvement is slight (**70% to 73%** on seen objects, **35% to 50%** on unseen objects), because the real dataset is small and highly skewed, constraining how much latent steering can correct. In contrast, pretraining RFS in simulation exposes a much broader range of grasps, poses, and perturbations. This diversity allows RFS to learn generalizable adaptation strategies that transfer effectively when finetuned on limited real data, leading to substantially larger improvements.
> We have now recorded the initial latent noise. We have included this information in the paper in **Section 6.2**.
>
> ---
>
> **Q: Clarification on Gradient Flow Through the Push ODE Module**
> >The Push operation is the standard denoising / ODE integration step used in diffusion and flow models. In our RL setting, Push is treated purely as a non-differentiable action-transformation module, part of the environment dynamics rather than the policy. In all experiments, Push uses five denoising steps.
>
> ---
>
> **Q: Clarification of Data Sources and Training Procedures in Simulation**
> >We thank the reviewer for pointing out the ambiguity. The VR-teleoperated demonstrations in **Section 5.1** are entirely simulation data collected within the simulator via a VR interface. In simulation, the high-level RFS policy is finetuned with online PPO, and the VR demonstrations are used only to pretrain the diffusion/flow policy. No real-world data is involved in the simulation experiments.
> We have revised **Section 5.1** to make explicit that (i) the demonstrations are simulation-generated VR data, and (ii) simulation finetuning uses online RL, clearly distinguishing this pipeline from the real-robot offline-RL setting.
>
> ---
>
> We appreciate your insightful feedback. Please let us know if there are any additional questions we can address.

---

> > ### Comment · Reviewer_kxpu · 2025-11-25
> >
> > Thank the authors for including additional experimental results and revising the paper.
> > For the question “separating the loss into two terms” and “detaching one branch when computing gradients for the other”, I mean using two losses, one for updating the $\pi_H$ and one for updating $\pi_r$. For example, when calculating the loss for $a_r$, detaching the gradient with respect to $a_0$. I am just curios if this will stabilize the training.
> > To further improve the paper, I encourage the authors to reorganize the Section 4 and Section 5 to combine them into a concise Methodology section and move some details to the Experiments section.
> > Considering the additional experimental results and the revision of the whole paper, I would like to increase the score to 6.

---

### Official Review · Reviewer_oCuu · 2025-10-28

**Soundness:** 2
**Presentation:** 2
**Contribution:** 3
**Rating:** 4
**Confidence:** 2

**Summary:**

The paper focuses on offline-to-online reinforcement learning (RL) setting.
Specifically, the paper proposes to adapt generative policies with reinforcement learning by unifying residual policy learning and diffusion steering into a new class of algorithms, residual flow steering (RFS).
The paper illustrates, through simulated and real-life experiments, that RFS outperforms compared algorithms.

**Strengths:**

- The unification between residual RL and latent-noise steering is interesting, and can open up avenues for various choices of $f$ and $g$.
- In the offline RL setting, RFS appears to generalize to unknown objects better in the real-life setting. The extra robustness experiments are helpful in demonstrating RFS' benefits.

**Weaknesses:**

- In section 5.2, the paper proposes to collect extra human correction data $a$---this seems to be a strong limitation, similar to applying the DAgger algorithm. It is possible that I have totally misunderstood this process:
	- The whole trajectory $((o_1, s_1), (o_2, s_2), \dots)$ is generated using the correction action $a$, as opposed to below.
	- First sample the trajectory $((o_1, s_1), (o_2, s_2), \dots)$ using the base policy actions, then obtain the correction actions based on the already collected trajectory.
	- Nevertheless, perhaps the paper can clarify this on lines 296-298.
- Experiments
	- The experimental setting is confusing. In section 5.2 it is mentioned that the RFS is finetuned via offline RL, but in the experiment, specifically section 6.1.2, the setting is now in the online RL setting.
		- In 6.1.2, does the learner obtain any human-correction data for finetuning, in addition to the PPO trajectories?
		- Secondly, if the setting is originally offline-to-online RL, I think it's unfair to compare the learning curve of Tabula-rasa RL, and instead it should be compared against algorithms such as Cal-QL, AWAC-like algorithms, or RLPD.
		- Instead, the rationale on the chosen baselines on action-space reduction and action codebooks are unclear. Is the intuition to reduce the sample efficiency through easier exploration with smaller action space?
	- Likewise, for section 6, the choice of compared algorithms can be strengthened with offline RL algorithms like CQL and IQL.

**Questions:**

See above.

---

> ### Author Response · Authors · 2025-11-25
> **Official Comment by Authors**
>
> Dear Reviewer,
>
> Thank you for your feedback on how we can improve our paper. We respond to individual comments below:
>
> ---
> **Q: Baselines Coverage**
> > We expanded our evaluation to include the offline-to-online RL baselines (AWAC, IQL, Flow Q-Learning) and the RL-with-demonstrations baselines (RLPD, IBRL) shown in **Table 1**. Across these methods, performance remains limited: offline-to-online RL reaches **22–55%**, and RL with demonstrations plateaus at **40–55%**, as both remain tightly constrained by the demonstration distribution and cannot adapt global behavior through online interaction. For completeness, we also report diffusion/flow RL finetuning, including DPPO and Reinflow (**40–60%**), and state-of-the-art residual RL, including Policy Decorator and ResiP (**50–70%**), which likewise cannot achieve broad behavioral adaptation. In contrast, **RFS reaches 97%**, showing that existing methods are restricted to either demonstration-anchored policies or local corrections, whereas RFS provides the global adaptation required for high performance.
> ---
> **Q: Choice of PCA and VQ-ACE baselines**
> >We include PCA[1] and VQ-ACE[2] because they are standard action-space reduction baselines for simplifying exploration in high-DoF dexterous control. Both of these methods reduce or discretize the action space, ensuring that all sampled actions remain in a more reasonable subspace and reducing the number of random actions. RFS is related because it also changes the action representation, but in a different way. Instead of compressing the action space via dimensionality reduction, RFS reshapes it into a more structured form by using latent noise for global strategy changes and residual actions for precise local adjustments. These baseline comparisons show that simple dimensionality (PCA/VQ-ACE) is not as effective as using exploration in RFS’s more structured action space.
>
> >[1] Haoqi Yuan, Bohan Zhou, Yuhui Fu, and Zongqing Lu. *Cross-embodiment dexterous grasping with reinforcement learning*, ICLR 2025
> >[2] Chenyu Yang, Davide Liconti, and Robert K. Katzschmann. *VQ-ACE: Efficient policy search for dexterous robotic manipulation via action chunking embedding*, arXiv 2024.
> ---
> **Q: Clarification of offline data collection:**
> > To clarify, our real-world finetuning does not use DAgger-style interactive correction. We collect a small offline teleoperation dataset once: the base policy executes the trajectory, and the human provides a corrective action only at the visited state. Each sample is simply (o, s, $a_{\text{base}}$, $a_{\text{human}}$, o′, s′), from which we compute the residual $a_r = a_{\text{human}} - a_{\text{base}}$. No iterative expert rollouts, on-policy supervision, or dataset aggregation are required. We have rephrased this in **Section 5.2** of the paper.
> ---
> **Q: Clarification of the Experimental Setting:**
> >The paper uses two different finetuning regimes, depending on whether experiments are run in simulation or on the real robot:
>
> >**Simulation (Section 6.1, including 6.1.2):**
> >- RFS is finetuned with online RL (PPO) using simulated interactions, starting from human-demonstration trajectories collected in VR.
> >- No human-correction data is used in the simulation.
>
> >**Real world (Section 5.2):**
> >- RFS is finetuned with offline RL using a small set of human corrective demonstrations collected on hardware.
> >- There is no online RL on the real robot performed at this time.
> ---
> **Paper modification**
> >We have revised Sections **5.1/5.2 and 6** for clarity and added an overview figure (**Figure 4**) that explicitly summarizes the two settings.
> ---
>
> We appreciate your insightful feedback. Please let us know if there are any additional questions we can address.

---

> > ### Comment · Reviewer_oCuu · 2025-11-27
> >
> > Thank you for the clarification and the additional experiments. I have increased the score.

---

### Official Review · Reviewer_yEFi · 2025-10-31

**Soundness:** 3
**Presentation:** 2
**Contribution:** 2
**Rating:** 6
**Confidence:** 3

**Summary:**

The paper considers the problem of fine-tuning pretrained generative (flow matching) policies for reinforcement learning. The contribution of this work is to introduce a general framework that incorporates two types of preexisting fine-tuning methods: input modulation (flow steering) and output modulation (residual learning). The authors propose a specific instance of this framework: Residual Flow Matching (RFS) which learns both an initial latent noise distribution and a policy that outputs residual actions, which is added to the push forward action produced by the pretrained model (pushed forward from the learned noise). The authors conduct simulated and real-life experiments in the dextrous manipulation robotics domain, and the experiments demonstrate RFS achieving favorable performance.

**Strengths:**

1. The paper integrates two previous methods (flow steering, residual learning) with complementary benefits and drawbacks to get the best features of both.
2. The framework the paper introduces is broad enough to be applicable to many important reinforcement learning applications, even outside the domain of manipulation, especially given the wide adoption of generative policies in various RL applications.

**Weaknesses:**

1. **Some Baseline choices lack motivation/clarity:**  I do not understand the use of the VQ-VAE and PCA baselines in Section 6.1.2/Table 1/Fig. 4. From my understanding, both are methods to get different state-action representations, whereas the focus of the paper is finetuning. If my understanding is correct, these results are comparing PPO finetuned with RFS with PPO trained for two separate state-action representations (made by PCA, VQ-VAE). Why is such a comparison meaningful?

2. **Seeds for experiments:** For the simulation experiments, it is really not valid to make claims based on only 3 random seeds, especially given that the paper is mostly empirical in nature.

3. Unless I am mistaken, residual RL is not a baseline for `w/ hand` in Table 1. Given that the authors propose a method that is essential residual RL+flow steering, I feel residual RL should be included for the `w/ hand` comparisons.

**Questions:**

Besides addressing my points in the weaknesses section, please make the following changes:
1. The notation for math in the paper is a bit cluttered. Specifically, please do not use the symbol $a$ whenever you write $a=a_r+a_b$, as we also have $\pi(a|s)$ (for example in Equation 4).
2. Grammatical mistakes:

a. In the abstract the sentence "Doing so allows policies to perform both local (residual actions) and
global exploration (latent noise), data-efficient adaptation." seems grammatically incorrect.

b. Please revise In table 1: absolute joint psose -> absolute joint pose.

---

> ### Author Response · Authors · 2025-11-24
> **Official Comment by Authors**
>
> Dear Reviewer,
>
> Thank you for your feedback on how we can improve our paper. We respond to individual comments below:
>
> ---
> **Q: Baseline for residual RL with hand**
> >We have added the baseline for residual RL by hand. This baseline achieves a success rate of **0.740 ± 0.05** (Table 1), compared with RFS, which reaches **0.971 ± 0.01**. While residual RL provides helpful local corrections, it is limited to refining the actions of the pretrained policy and cannot modify the global manipulation strategy.
> We have also added nine baselines spanning flow-RL finetuning, offline-to-online RL, RL with demonstrations and residual RL (Table 1). Their performance ranges from **22% to 74%**, compared to 97% for RFS. Flow-RL finetuners reach **41–58%**, residual RL **52–74%**, and offline/online RL or demo-based methods **22–55%**. It further demonstrates that RFS is capable of both effective global adaptation and local refinement, which neither of the baseline methods achieves.
> ---
> **Q: The motivation of VQ-ACE and PCA baselines**
> > We include PCA[1] and VQ-ACE[2] because they are standard action-space reduction baselines for simplifying exploration in high-DoF dexterous control. Both of these methods reduce or discretize the action space, ensuring that all sampled actions remain in a more reasonable subspace and reducing the number of random actions. RFS is related because it also changes the action representation, but in a different way. Instead of compressing the action space via dimensionality reduction, RFS reshapes it into a more structured form by using latent noise for global strategy changes and residual actions for precise local adjustments. These baseline comparisons show that simple dimensionality (PCA/VQ-ACE) is not as effective as using exploration in RFS’s more structured action space.
>
> >[1] Haoqi Yuan, Bohan Zhou, Yuhui Fu, and Zongqing Lu. *Cross-embodiment dexterous grasping with reinforcement learning*, ICLR 2025
> >[2] Chenyu Yang, Davide Liconti, and Robert K. Katzschmann. *VQ-ACE: Efficient policy search for dexterous robotic manipulation via action chunking embedding*, arXiv 2024
>
> ---
> **Q: Writing mistakes**
> >We have updated the paper to address the reviewers’ concerns: we corrected several unclear mathematical expressions (**Section 5**), improved the formatting of the equations (**Section 5**), and corrected grammar issues throughout the abstract and main text.
> ---
> **Q: Seeds for experiments**
> >We have run all our updated experiments with five random seeds.
> ---
> We appreciate your insightful feedback. Please let us know if there are any additional questions we can address.

---

### Official Review · Reviewer_22ER · 2025-11-01

**Soundness:** 3
**Presentation:** 3
**Contribution:** 2
**Rating:** 2
**Confidence:** 3

**Summary:**

The paper proposes Residual Flow Steering (RFS), an RL fine‑tuning scheme for flow‑matching / diffusion policies that jointly (i) steers the initial latent noise for global changes and (ii) applies a residual action for local corrections—without modifying base policy parameters. The method is instantiated for dexterous grasping: pretrain in simulation (Leap Hand + Franka), distill a visuomotor policy, then fine‑tune in the real world with offline RL (TD3+BC). Experiments show improved success over action‑representation baselines and ablations (residual‑only or steering‑only) in both pinch and power grasps.

**Strengths:**

RFS is a clean unification of residual RL (output modulation) and latent steering (input modulation), formalized via a modulation policy.

Strong, consistent improvements over baselines and over residual‑only / steering‑only ablations in simulation and better real‑world success vs. zero‑shot and supervised fine‑tuning.

The paper is well written. The introduction and the method are easy to follow and communicate the contributions and the implementation well.

**Weaknesses:**

Limited novelty. The algorithm mainly combines two widely used adaptation strategies (residual action learning and latent steering).

Baseline coverage. Real‑world evaluation lacks comparisons to other RL fine‑tuning approaches for diffusion/flow policies (e.g., recent flow‑RL fine‑tuners); most comparisons are ablations or action‑space baselines.

Task scope. Validation centers on grasping; the paper claims broader applicability, but no additional manipulation tasks are shown.

**Questions:**

Why is “changing the action representation” (absolute/relative joints, PCA, VQ‑VAE) considered a primary baseline for the first task? What does that comparison communicate about the contribution of the paper?

Could you add head‑to‑head comparisons against other RL fine‑tuning methods for diffusion/flow policies to strengthen the claim that RFS is preferable beyond its own ablations?

Do you have preliminary results (even in sim) on more complex tasks (e.g., non‑prehensile skills or multi‑stage manipulation) to support generalization claims?

---

> ### Author Response · Authors · 2025-11-24
> **Official Comment by Authors**
>
> Dear Reviewer,
>
> Thank you for your feedback on how we can improve our paper. We respond to individual comments below:
>
> ---
>
> ### **Q: Limited novelty**
>
> > While RFS does indeed build on residual RL and latent-noise steering, we argue that this combination of these two techniques is a non-trivial scientific study. The reasoning here is two-fold:
> > **1)** RFS leads to a very sizable empirical improvement over a large variety of baselines, and significantly better than both residual RL and diffusion steering. RFS unifies these two modulation pathways into a single RL objective, enabling them to interact and acquire adaptation capabilities that neither approach can achieve on its own.
> > **2)** The design choices made in RFS (for instance, the choice of representation of actions in a Q function, Section 5.2 and 6.2) make a sizable difference in performance.
>
> ---
>
> ### **Q: Baseline coverage**
>
> > As suggested, we have significantly expanded our set of baseline comparisons. We have added **9 new baselines**, including two RL finetuning methods for flow policies (DPPO, ReinFlow), offline-to-online RL (AWAC, IQL, Flow Q-Learning), RL with demonstrations (RLPD, IBRL), and SOTA residual RL (Policy Decorator, ResiP) (Table 1; Sec. 6.1.1).
> > For diffusion/flow RL finetuning, DPPO and ReinFlow together reach **41–58%**, since online RL can explore only indirectly through the pretrained generative model. For other baselines, their performances span **22–55%** (offline/online and RL with demonstrations) and **52–74%** (residual RL), all far below **RFS’s 97%**. These results demonstrate that RFS’s unified latent-steering and residual-refinement design enables an adaptation capability not achievable by existing approaches.
>
> ---
>
> ### **Q: Task scope**
>
> > As suggested, we have expanded our evaluation to include additional manipulation tasks. To start, we have added a pick-and-place task (**Appendix A.1**). On this problem, RFS achieves **0.939 ± 0.012**, substantially higher than all baseline methods (**Appendix A.1, Table 3**).
> > The updated results can also be visualized at:
> > **[https://residualflowsteering-png.github.io/residualflowsteering-png/rebuttal.html](https://residualflowsteering-png.github.io/residualflowsteering-png/rebuttal.html)**
> >
> > Diffusion/flow finetuners (DPPO, ReinFlow) reach only **0.433** and **0.462**; offline-to-online RL methods (IQL, AWAC, Flow Q-Learning) peak at **0.560**, **0.240**, and **0.486**; residual RL methods (Policy Decorator, ResiP, DSRL) reach **0.314** and **0.570**; and RL-with-demonstrations methods top out at **0.763**. Across all categories, RFS achieves the highest performance.
> >
> > We are currently extending our evaluation beyond this to non-prehensile manipulation (pushing) and long-horizon tasks (packing), which will be incorporated soon (before the end of the rebuttal period).
>
> ---
>
> ### **Q: The choice of VQ-ACE and PCA baselines**
>
> > We include PCA [1] and VQ-ACE [2] because they are standard action-space reduction baselines for simplifying exploration in high-DoF dexterous control. Both of these methods reduce or discretize the action space, ensuring that all sampled actions remain in a more reasonable subspace and reducing the number of random actions.
> >
> > RFS is related because it also changes the action representation, but in a different way. Instead of compressing the action space via dimensionality reduction, RFS reshapes it into a more structured form by using latent noise for global strategy changes and residual actions for precise local adjustments. These baseline comparisons show that simple dimensionality reduction (PCA/VQ-ACE) is not as effective as using exploration in RFS’s more structured action space.
>
> **References**
> [1] Haoqi Yuan, Bohan Zhou, Yuhui Fu, and Zongqing Lu. *Cross-embodiment dexterous grasping with reinforcement learning*, ICLR 2025.
> [2] Chenyu Yang, Davide Liconti, and Robert K. Katzschmann. *VQ-ACE: Efficient policy search for dexterous robotic manipulation via action chunking embedding*, arXiv 2024.
>
> ---
>
> We appreciate your insightful feedback. Please let us know if there are any additional questions we can address.

---

### Author Response · Authors · 2025-11-24
**General Responses and Summary of Updates**

We thank the reviewers for the detailed feedback. As per reviewer feedback,we have substantially  **expanded our set of baseline comparisons**, and we now cover most of the baselines suggested by the reviewers. We outline our additional experiments below, and the detailed analysis and comparison results can be found in **Table 1** and **Section 6.1.1** of the updated PDF:



- **Diffusion/flow finetuning:**   **DPPO**[3], **ReinFlow**[4], suggested by Reviewers 22ER and kxpu
  > **Result:** DPPO finetuning slightly degrades performance (**45% → 41%**), and ReinFlow improves only modestly (**47% → 58%**), whereas **RFS reaches up to 97% (>40% relative gain)** (Table 1; Sec. 6.1.1).  We find that only adjusting initial noise and residual terms (as in RFS) can be more stable than performing RL through the entire denoising chain (as in DPPO/ReinFlow).



- **Offline to online RL:**  **IQL**[5], **AWAC**[6], **Flow Q-Learning**[7], suggested by Reviewer oCuu
  > **Result:** Offline performance plateaus at **22–55%** for these baselines for offline-to-online RL (Table 1; Sec. 6.1.1).  We find that offline-to-online RL methods often don’t explore in directed ways, while RFS leverages base-policy pretraining for directed exploration.



- **RL with demonstrations:**  **RLPD**[8], **IBRL**[9], suggested by Reviewer oCuu
  > **Result:** These methods improve exploration over learning from scratch, but **plateau at 40–55%** (Table 1; Sec. 6.1.1).  By performing input/output modulation on the base policy, RFS is able to retain the base-policy performance and continue to improve.



- **Residual RL:**  **Policy Decorator**[10], **ResiP**[11], suggested by Reviewers kxpu and yEFi
  > **Result:** Residual RL outperforms previous baselines (**52–74%**, Table 1; Sec. 6.1.2), but lacks global behavior modulation. RFS improves further to 94–97%, since it can modulate global exploration via input latent-noise selection.

---

In addition, we include **PCA**[1] and **VQ-ACE**[2] because they are standard action-space reduction baselines in dexterous RL, commonly used to improve sample efficiency and simplify exploration by compressing the high-DoF action space.
Given that RFS effectively modifies the action space to explore in a more “meaningful” latent space, these are appropriate comparisons.

Furthermore, we have revised the **related work section** to include these additional baseline methods and to provide a broader overview of prior approaches.
In addition, Sections **4 and 5** have been updated to improve clarity, and **Section 6** has been updated with the above-mentioned baseline comparisons.

Reviewers can also refer to their review-specific responses, where we address the individual concerns in detail.

---

### **References**

[1] Haoqi Yuan, Bohan Zhou, Yuhui Fu, and Zongqing Lu. *Cross-embodiment dexterous grasping with reinforcement learning*, ICLR 2025.

[2] Chenyu Yang, Davide Liconti, and Robert K. Katzschmann. *VQ-ACE: Efficient policy search for dexterous robotic manipulation via action chunking embedding*, arXiv 2024.

[3] Allen Z. Ren, Justin Lidard, Lars L. Ankile, Anthony Simeonov, Pulkit Agrawal, Anirudha Majumdar, Benjamin Burchfiel, Hongkai Dai, and Max Simchowitz. *Diffusion Policy Policy Optimization*, ICLR 2025.

[4] Tonghe Zhang, Chao Yu, Sichang Su, and Yu Wang. *ReinFlow: Fine-tuning Flow Matching Policy with Online Reinforcement Learning*, NeurIPS 2025.

[5] Ilya Kostrikov, Ashvin Nair, and Sergey Levine. *Implicit Q-Learning*, ICLR 2022.

[6] Ashvin Nair, Abhishek Gupta, Murtaza Dalal, and Sergey Levine. *AWAC: Accelerating Online Reinforcement Learning with Offline Datasets*, arXiv 2021.

[7] Seohong Park, Qiyang Li, and Sergey Levine. *Flow Q-learning*, ICML 2025.

[8] Philip J. Ball, Laura Smith, Ilya Kostrikov, and Sergey Levine. *Efficient Online Reinforcement Learning with Offline Data*, ICML 2023.

[9] Hengyuan Hu, Suvir Mirchandani, and Dorsa Sadigh. *Imitation Bootstrapped Reinforcement Learning*, RSS 2024.

[10] Xiu Yuan, Tongzhou Mu, Stone Tao, Yunhao Fang, Mengke Zhang, and Hao Su. *Policy Decorator: Model-Agnostic Online Refinement for Large Policy Model*, ICLR 2025.

[11] Lars Ankile, Anthony Simeonov, Idan Shenfeld, Marcel Torne, and Pulkit Agrawal. *From Imitation to Refinement – Residual RL for Precise Assembly*, ICRA 2025.

---

### Author Response · Authors · 2025-12-03
**Summary of Updates by Authors**

Dear AC,

Below is a consolidated summary of our rebuttal to help you quickly navigate the main concerns raised across reviewers. For full details, please refer to our general response and the reviewer-specific replies. We also note that Reviewers kxpu and oCuu acknowledged our clarifications and have raised their scores to 6. We believe we have added a considerable number of experiments and revisions to address all the major concerns raised by reviewers.

**Task Scope**: suggested by Reviewer 22ER

> We expanded our evaluation beyond grasping by adding three additional settings—**pick-and-place, push-to-grasp, and long-horizon packing**. All baselines and RFS are trained with five random seeds. All new experiments and baseline comparisons are included in **Section 5 (updated Table 1 and Figure 6)**.
>
>
> For convenience, we also provide a visual summary of the updated results at:
>
> https://residualflowsteering-png.github.io/residualflowsteering-png/rebuttal.html
>
> **Overall, RFS improves over the base policy and achieves the highest success rates across all baselines: 0.89 (grasp), 0.94 (pick-place), 0.78 (packing), and 0.72 (push-to-grasp)**.
>

**Baseline Coverage**: suggested by Reviewers 22ER, oCuu, and kxpu

> Per reviewer feedback, we have run the suggested baselines—adding nine additional methods spanning **diffusion/flow RL fine-tuning, offline-to-online RL, RL with demonstrations, and the SOTA residual RL methods**. Across all tasks, RFS continues to outperform every baseline significantly. A summary is provided below, with full results in Table 1 and Figure 6 (Sec. 5).
>
> - **Diffusion/flow finetuning:** DPPO[3], ReinFlow[4], suggested by Reviewers 22ER and kxpu
>
>     **Result:** DPPO underperforms across all tasks (**all < 0.50**). ReinFlow is slightly stronger—**0.58 (grasp), 0.46 (pick-place), 0.39 (packing)**—but still fails on **push-to-grasp (0.10)**. We find that only adjusting the initial noise and residual terms (as in RFS) is more stable (**achieving success rates at least 0.35 higher on each task**) than performing RL across the entire denoising chain (as in DPPO/ReinFlow).
>
> - **Offline → online RL**: IQL[5], AWAC[6], Flow Q-Learning[7], suggested by Reviewer oCuu
>
>     **Result:** IQL is the strongest offline-to-online baseline—**0.69 (grasp), 0.56 (pick-place)** —but still struggles on **packing (0.20) and push-to-grasp (0.26)** , while AWAC and FlowQ **collapse** on multiple tasks. We find that offline-to-online RL methods often don’t explore in a directed way, whereas RFS tends to leverage base-policy pre-training for directed exploration, **yielding ~0.20 higher success on grasping and pick-and-place and ~0.50 higher on push-to-grasp and packing across all offline-to-online RL baselines** .
>
> - **RL with demonstrations:** RLPD[8], IBRL[9], suggested by Reviewer oCuu
>
>     **Result:** RLPD performs well on **grasping (0.54) and pick & place (0.76)**, but **collapses** on long-horizon ones, while IBRL works only on the packing and grasping task and fails elsewhere. By modulating input/output only on the base policy, RFS can retain the base policy's performance while further **improving to >0.70 on packing and push-to-grasp and ~0.90 on grasping and pick-and-place** .
>
> - **Residual RL:** Policy Decorator[10], ResiP[11], suggested by Reviewers kxpu and yEFi
>
>     **Result:** Policy Decorator and ResiP outperform the above baselines across most tasks. However, their improvements (**~0.60 for grasping, 0.50 for pick & place** ) remain local, and they struggle with tasks that require broader, sequence-level adaptation, such as **packing (0.40) and push-to-grasp (0.15).** By comparison, RFS attains **over 0.70 on packing and push-to-grasp and approximately 0.90 on grasping and pick-and-place** .
>
> - **DSRL[12]**
>
>     **Result:** DSRL is the strongest baseline, achieving **0.74 (grasp), 0.70 (pick-place), 0.64 (packing), and 0.43 (push-to-grasp)** . But it cannot reliably generate corrections outside the demonstration distribution, limiting its ability to adapt to challenging or out-of-distribution states, whereas **RFS can reach 0.89 (grasping), 0.94 (pick-and-place), 0.78 (packing), and 0.72 (push-to-grasp)** .
>

**Clarifying PCA & VQ-ACE Baselines**: suggested by all reviewers

> We include PCA[1] and VQ-ACE[2] because they are standard action-space reduction baselines in dexterous RL, commonly used to **improve sample efficiency and simplify exploration by compressing the high-DoF action space**. Given that RFS effectively modifies the action space to explore a more “meaningful” latent space, these comparisons are appropriate.
>

---

> ### Author Response · Authors · 2025-12-03
> **Summary of Updates by Authors**
>
> **Applying RFS to Real-Only Demonstration Policies**:suggested by reviewer kxpu
>
> > We evaluated RFS when applied directly to a policy trained only on real-world demonstrations, without any simulation pre-training. In this setting, the improvements are relatively modest (**70%→73% on seen; 35%→50% on unseen**). This can be attributed to the small and skewed real-world dataset, which limits the impact of latent steering. Simulation pre-training provides a broader grasp and perturbation coverage, with the broader base policy coverage allowing for more effective RFS adaptation in the real world. Related details are now included in Section 5.2.
> >
>
> **Novelty of the RFS**:suggested by reviewer 22er
>
> > While RFS builds on residual RL and latent-noise steering, it is not a simple combination. Latent steering cannot go beyond the base policy’s behavior. In contrast, RFS, by jointly modulating inputs and outputs under a single RL objective, enables adaptation beyond the base policy and suggests yields substantial gains—**+0.15–0.35 on grasping/pick-place** and **+0.30–0.60 on packing/push-to-grasp over all baselines**. In addition, crucial design choices (e.g., action representation in the Q-function; Sec. 4.2.2, 5.3) are essential for stability and performance, underscoring that RFS is a principled algorithmic contribution rather than a superficial merge.
> >
>
> **Manuscript Revisions**
>
> > Reviewers also raised several smaller, reviewer-specific misconceptions (e.g., the “Push” operation, simulation and real-world setup, data source and training procedures in simulation and offline data collection). We addressed each of these in detail in the reviewer-specific responses. In the paper, we significantly expanded **Section 2** (related work) by including many of the papers listed below, clarified the methodology in **Section 4**, and updated the results and analysis in **Section 5**. All changes made during the rebuttal are marked in blue text to make them easy to identify.
> >
>
> **Clarification on experiment setup**
>
> > To address Reviewer oCuu’s misunderstanding about the experiment setup and the offline data collection process in the real world, we revised **Sec. 4.2 and added Fig. 1**. As shown in Fig. 1, RFS is fine-tuned online in simulation from VR demos, and fine-tuned offline in the real world using a small set of human corrective demonstrations collected on hardware, where the base policy runs the trajectory and the human provides corrections only at visited states. Reviewer oCuu acknowledged our clarifications and has raised the scores to 6.
> >
>
> [1] Haoqi Yuan, Bohan Zhou, Yuhui Fu, and Zongqing Lu. Cross-embodiment dexterous grasping with reinforcement learning, ICLR 2025.
>
> [2] Chenyu Yang, Davide Liconti, and Robert K. Katzschmann. Vq-ace: Efficient policy search for dexterous robotic manipulation via action chunking embedding, arXiv 2024
>
> [3] Allen Z. Ren, Justin Lidard, Lars L. Ankile, Anthony Simeonov, Pulkit Agrawal, Anirudha Majumdar, Benjamin Burchfiel, Hongkai Dai, and Max Simchowitz. Diffusion policy policy optimization, ICLR 2025
>
> [4] Tonghe Zhang, Chao Yu, Sichang Su, and Yu Wang. Reinflow: Fine-tuning flow matching policy with online reinforcement learning, NeurIPS 2025
>
> [5] Ilya Kostrikov, Ashvin Nair, and Sergey Levine. Offline reinforcement learning with implicit q- learning, ICLR 2022
>
> [6] Ashvin Nair, Abhishek Gupta, Murtaza Dalal, and Sergey Levine. Awac: Accelerating online reinforcement learning with offline datasets, arXiv 2021
>
> [7] Seohong Park, Qiyang Li, and Sergey Levine. Flow Q-learning, ICML 2025
>
> [8] Philip J. Ball, Laura Smith, Ilya Kostrikov, and Sergey Levine. Efficient online reinforcement learning with offline data, ICML 2023
>
> [9] Hengyuan Hu, Suvir Mirchandani, and Dorsa Sadigh. Imitation bootstrapped reinforcement learning, RSS 2024
>
> [10] Xiu Yuan, Tongzhou Mu, Stone Tao, Yunhao Fang, Mengke Zhang, and Hao Su. Policy decorator: Model-agnostic online refinement for large policy model, ICLR 2025
>
> [11] Lars Ankile, Anthony Simeonov, Idan Shenfeld, Marcel Torne, and Pulkit Agrawal. From imitation to refinement – residual rl for precise assembly, ICRA 2025
>
> [12] Andrew Wagenmaker, Mitsuhiko Nakamoto, Yunchu Zhang, Seohong Park, Waleed Yagoub, Anusha Nagabandi, Abhishek Gupta, and Sergey Levine. Steering your diffusion policy with latent space reinforcement learning, CORL 2025

---

### Meta-Review · Area_Chair_Sg3d · 2026-01-07

**Summary:**

This paper proposes Residual Flow Steering (RFS), a framework for adapting pretrained flow-matching policies via joint latent-noise steering and residual action correction. Reviewers found the approach technically sound and the core idea intuitive, but raised concerns regarding novelty, baseline coverage, task scope, and clarity of the experimental setup. The authors’ rebuttal is extensive and directly addresses many of these concerns, adding numerous new baselines, tasks, clarifications, and analyses. Notably, two reviewers who initially gave negative evaluations (score 4) explicitly stated that they would increase their scores following the rebuttal. While I recommend acceptance, I want to explicitly note that the revised version of the paper includes an unusually large number of changes, including substantial new experiments and results, which is materially different from the originally reviewed version.

**Reviewer Concerns:**

The rebuttal substantially addresses several major reviewer concerns, particularly by expanding baseline coverage to include diffusion/flow RL fine-tuning methods, offline-to-online RL, RL with demonstrations, and state-of-the-art residual RL, as well as by broadening the task scope beyond grasping to include pick-and-place, push-to-grasp, and long-horizon packing with increased experimental rigor. The authors also clarified previously confusing aspects of the experimental setup, data collection, and training pipeline, resolving misunderstandings raised by reviewers. However, some concerns remain only partially addressed: while the authors strengthen the novelty argument and demonstrate clear empirical gains, the core contribution still rests on a unification of existing techniques rather than a fundamentally new algorithm.

**Reviewer Scores:**

Two of the reviewers with an initial borderline rejection review explicitly indicated in comment that they would increase the score during the rebuttal. The reviewer that had concern over novelty is less likely to update their score.

---

### Decision · Program_Chairs · 2026-01-26

Accept (Poster)